# LDLCC: Label Distribution Learning-Based Confidence Calibration for Crowdsourcing

## Abstract

Crowdsourcing typically collects multiple noisy labels for each instance and then aggregates these labels to infer its unknown true label. We discover that miscalibration, an important issue in supervised learning, also frequently arises in label aggregation. Miscalibration prevents existing label aggregation methods from assigning accurate confidence when inferring aggregated labels. However, in downstream tasks of label aggregation, both the aggregated labels and their associated confidence are equally significant. To address this issue, we formally define confidence calibration for crowdsourcing and propose a novel Label Distribution Learning-based Confidence Calibration (LDLCC) method in this paper. Specifically, to mitigate the impact of noisy labels, we first identify high-confidence instances and sharpen their label distributions based on the results of label aggregation. Subsequently, to avoid the overconfidence caused by the translation invariance of softmax, we train a regression network to learn the label distribution of each instance. Finally, to obtain the calibrated confidence of each aggregated label, we normalize the learned distribution from the regression network and take its maximum value. Extensive experimental results indicate that LDLCC can serve as a universal post-processing method to calibrate the confidence of each aggregated label, and thus further enhance the performance of downstream tasks.

## 1 Introduction

Crowdsourcing provides an efficient and economical approach to obtaining large-scale annotated data, catering to the needs of data-hungry models in supervised learning (Jiang et al., 2022; Zhang, 2022). However, due to the poor expertise, the labels collected from crowd workers are noisy (Li et al., 2020). To mitigate the impact of noisy labels, crowdsourcing introduces a mechanism called *repeated labeling* (Sheng et al., 2008). Repeated labeling ensures that each instance is annotated by different workers to obtain multiple noisy labels. Subsequently, *label aggregation* is performed to aggregate these noisy labels to infer its unknown true label.

Currently, a large number of label aggregation methods have been proposed (Dawid & Skene, 1979; Sheng et al., 2019; Ying et al., 2024; Zhang et al., 2025). These methods primarily focus on improving the accuracy of aggregated labels, gradually narrowing the gap between aggregated labels and the unknown true labels. However, despite the effectiveness of these label aggregation methods, the labels they aggregate still inherently contain a certain degree of noise (Li et al., 2023a). This fact has driven the development of downstream tasks of label aggregation, such as noise correction (Zhang et al., 2018) and learning from noisy labels (Karim et al., 2022). For these downstream tasks, providing only aggregated labels is often insufficient, the confidence of each aggregated label is equally significant. Here, the confidence reflects how "close" or "far" an instance is to its aggregated label (e.g., 0.99 or 0.01). For example, in noise correction, if the confidence of an aggregated label is low, we usually tend to identify the corresponding instance as a noisy one. Conversely, if the confidence is high, we usually tend to identify the corresponding instance as a clean one.

Unfortunately, we discover that miscalibration frequently arises in label aggregation methods. Here, miscalibration refers to a mismatch between the confidence and the correctness of aggregated labels inferred by a label aggregation method. Take Majority Voting (MV) as an example (Sheng et al., 2008). If an instance $x$ receives only three labels and the values of them are $(c_1, c_1, c_2)$, MV will infer the aggregated label of $x$ as $c_1$ with a confidence value of 0.67. However, due to the presence

of noisy labels, the correctness of $x$ belonging to $c_1$ varies considerably from 0.67. Considering the miscalibration, it is essential to perform confidence calibration for label aggregation methods.

Although many calibration methods have been proposed in supervised learning, they typically rely on true labels (Guo et al., 2017; Mukhoti et al., 2020). However, in the crowdsourcing scenario we focus on, true labels of instances are unknown, and only aggregated labels are available. This makes calibration methods from supervised learning unreliable. To address this issue, we formally define confidence calibration for crowdsourcing in this paper. Subsequently, inspired by label distribution learning (Xu & Geng, 2019; Lu et al., 2023), we propose a novel Label Distribution Learning-based Confidence Calibration (LDLCC) method. Specifically, LDLCC first identifies high-confidence instances from all instances and sharpens their label distributions to mitigate the impact of noisy labels. Then, LDLCC trains a regression network to learn the label distribution of each instance to avoid the overconfidence caused by the translation invariance of softmax. Finally, LDLCC normalizes the learned distribution from the network and takes its maximum value to obtain the calibrated confidence of each aggregated label. In summary, the main contributions of this paper are as follows:

- We provide a formal definition of confidence calibration for crowdsourcing, which clarifies the differences from calibration in supervised learning and maximizes the utilization of information in crowdsourcing scenarios.

- We design a strategy to identify high-confidence instances based on the results of label aggregation. By sharpening the label distributions of high-confidence instances, we mitigate the impact of noisy labels.

- We propose a method called LDLCC to calibrate the confidence of aggregated labels. By training a regression network to learn the label distribution of each instance, we avoid the overconfidence caused by the softmax.

- We conduct extensive experiments to verify the effectiveness of our LDLCC. The results show that LDLCC can serve as a universal post-processing method to calibrate the confidence of each aggregated label.

## 2 RELATED WORK

**Label Aggregation.** Label aggregation methods can be divided into one-stage and two-stage methods. One-stage methods directly use crowd labels to train neural networks, and the predictions of the trained networks can serve as aggregated labels (Rodrigues & Pereira, 2018; Chen et al., 2020; Li et al., 2023b). The simplest two-stage method is Majority Voting (MV), which assigns the class with the highest vote count as the aggregated label (Sheng et al., 2008). Subsequently, numerous variants of MV have been proposed to improve its performance (Li & Yu, 2014; Tian et al., 2019; Chen et al., 2022). Another classic two-stage method is DS (Dawid & Skene, 1979), which optimizes the confusion matrices of workers and the aggregated labels of instances using the Expectation-Maximization (EM) algorithm. Raykar et al. (2010) and Kim & Ghahramani (2012) are Bayesian versions of DS, designed for binary and multi-class tasks, respectively. Recently, several methods based on the idea of nearest neighbors have been proposed (Jiang et al., 2022; Ying et al., 2024; Zhang et al., 2024; 2025). By leveraging information from neighboring instances or neighboring workers, these methods have improved the performance of label aggregation. However, neither one-stage nor two-stage methods directly address the issue of miscalibration.

**Downstream Tasks of Label Aggregation.** Noise correction and learning from noisy labels are two common downstream tasks of label aggregation. In noise correction, instances are usually divided into a clean set and a noisy set based on the confidence of the aggregated labels (Zhang et al., 2018; Xu et al., 2021). One or more models are then trained on the clean set to correct the instances in the noisy set (Li et al., 2023c; Su et al., 2026). Learning from noisy labels can be broadly divided into loss correction and example selection (Zong et al., 2024). Loss correction aims to correct the loss by estimating the noise transition matrix and adjusting the labels or weights of instances (Goldberger & Ben-Reuven, 2017; Shu et al., 2019). Example selection aims to identify clean instances from datasets and then perform semi-supervised learning by treating remaining instances as unlabeled instances (Huang et al., 2019; Karim et al., 2022). For the above methods, both the aggregated labels and their confidence play a vital role.

**Calibration in Supervised Learning.** In supervised learning, methods to address the issue of network miscalibration can be broadly divided into three categories (Tao et al., 2023b). The first category is post-hoc calibration methods, such as Histogram Binning (Zadrozny & Elkan, 2001) and Temperature Scaling (Guo et al., 2017), which adjust model predictions after training based on a held-out validation set. The second category is regularization-based calibration methods, such as Label Smoothing (Müller et al., 2019) and Weight Decay (Tao et al., 2023a), which achieve calibration by regularizing the input and target of networks, or directly ensembling different networks. The third category is loss-based calibration methods, such as Maximum Mean Calibration Error (Kumar et al., 2018) and Focal Loss (Mukhoti et al., 2020), which add a calibration term to the training loss or replace the training loss with other loss functions. Almost all these three categories of methods rely on true labels. However, true labels are unknown in crowdsourcing, which makes the above calibration methods cannot be directly applied to crowdsourcing.

## 3 PROBLEM FORMULATION

Considering a crowdsourcing task with $\mathcal{X}$ as the attribute space and $\mathcal{Y}$ as the label space, we define a crowdsourced dataset as $\mathcal{D} = \{(\boldsymbol{x}_i, \boldsymbol{L}_i)\}_{i=1}^N$. Here, $\boldsymbol{x}_i \in \mathcal{X}$ is the $i$-th instance in $\mathcal{D}$, which can be expressed as $\{x_{im}\}_{m=1}^M$. $M$ is the dimension of attributes, and $x_{im}$ denotes the attribute value of $\boldsymbol{x}_i$ on the $m$-th attribute $A_m$. $\boldsymbol{L}_i$ is the multiple noisy label set of $\boldsymbol{x}_i$, which can be expressed as $\{l_{ir}\}_{r=1}^R$. $R$ is the number of workers and $l_{ir}$ denotes the label of $\boldsymbol{x}_i$ annotated by the $r$-th worker $u_r$. $l_{ir}$ takes a value from a fixed set $\{-1, c_1, \ldots, c_q, \ldots, c_Q\}$, where $Q$ is the number of classes. $c_q \in \mathcal{Y}$ denotes the $q$-th class and $-1$ indicates that $u_r$ has not annotated $\boldsymbol{x}_i$. $y_i \in \mathcal{Y}$ is the true label of $\boldsymbol{x}_i$, which is unknown in crowdsourcing scenarios.

**Label Aggregation.** A label aggregation method can be expressed as $f : \mathcal{D} \to \mathcal{Y}$. Given a crowdsourced dataset $\mathcal{D}$, the label aggregation method $f$ first estimates a label distribution $\boldsymbol{P}_i$ over the label space $\mathcal{Y}$ for each instance $\boldsymbol{x}_i$. Subsequently, $f$ determines the aggregated label $\hat{y}_i$ based on $\boldsymbol{P}_i$. Typically, $\hat{y}_i$ is the class with the highest probability in $\boldsymbol{P}_i$, and thus the associated confidence of $\hat{y}_i$ is $\hat{p}_i = \max \boldsymbol{P}_i$. Existing label aggregation methods focus only on minimizing the error between $\hat{y}_i$ and $y_i$, neglecting the accuracy of $\hat{p}_i$, which ultimately leads to miscalibration. However, both $\hat{y}_i$ and $\hat{p}_i$ serve as inputs for downstream tasks of label aggregation and play important roles. Considering that inaccurate $\hat{p}_i$ will harm the performance of downstream tasks, we propose confidence calibration for crowdsourcing in this paper.

**Confidence Calibration.** We define confidence calibration as a downstream task of label aggregation as well, but it is performed prior to noise correction and learning from noisy labels. Once label aggregation is completed, the crowdsourced dataset $\mathcal{D}$ can be converted to $\hat{\mathcal{D}} = \{(\boldsymbol{x}_i, \boldsymbol{P}_i, \hat{y}_i, \hat{p}_i)\}_{i=1}^N$, which is used as the input for confidence calibration. Referring to the definition of network calibration in supervised learning (Guo et al., 2017), confidence calibration aims to ensure that the calibrated confidence $\tilde{p}_i$ accurately represents the true probability of the aggregated label $\hat{y}_i$ being correct. Formally, the perfectly calibrated confidence satisfies:

$$P(\hat{y}_i = y_i \mid \tilde{p}_i = p) = p, \quad \forall p \in (0, 1] \tag{1}$$

Similarly, we apply the Expected Calibration Error (ECE) to evaluate the performance of confidence calibration. Given the calibrated confidence $\tilde{p}_i$, we define the ECE as:

$$\text{ECE} = \mathbb{E}_{\tilde{p}_i} |P(\hat{y}_i = y_i \mid \tilde{p}_i) - \tilde{p}_i|. \tag{2}$$

In reality, the probability $P(\hat{y}_i = y_i \mid \tilde{p}_i)$ cannot be accurately estimated due to the finite instances in $\hat{\mathcal{D}}$. Therefore, an approximation of ECE is introduced. Specifically, we can separate all instances into $T$ bins $\{B_t\}_{t=1}^T$, where $B_t$ contains all the instances whose calibrated confidence $\tilde{p}_i \in (\frac{t-1}{T}, \frac{t}{T}]$. Subsequently, we can calculate the average confidence $\bar{p}_t = \frac{1}{|B_t|} \sum_{\boldsymbol{x}_i \in B_t} \tilde{p}_i$ and the accuracy $a_t = \frac{1}{|B_t|} \sum_{\boldsymbol{x}_i \in B_t} \mathbb{I}(\hat{y}_i = y_i)$ for each bin $B_t$. Here, $\mathbb{I}(\cdot)$ is the indicator function, which returns 1 if the condition is true and 0 otherwise. Finally, the approximated ECE can be calculated as follows:

$$\text{ECE} = \sum_{t=1}^T \frac{|B_t|}{N} |a_t - \bar{p}_t|, \tag{3}$$

where $|B_t|$ is the number of instances in $B_t$. It is worth noting that the approximated ECE can only be used in experiments and cannot be applied to designing confidence calibration methods because the true label $y_i$ is unknown.

**Differences and Challenges.** According to the definition of confidence calibration, we can see that it is different from calibration in supervised learning. The input of calibration in supervised learning is $\mathcal{D} = \{(\boldsymbol{x}_i, y_i)\}_{i=1}^N$, while the input of confidence calibration for crowdsourcing is $\hat{\mathcal{D}} = \{(\boldsymbol{x}_i, \boldsymbol{P}_i, \hat{y}_i, \hat{p}_i)\}_{i=1}^N$. This difference poses significant challenges to the confidence calibration of crowdsourcing. On the one hand, the true label $y_i$ is unknown in crowdsourcing, which makes it difficult to directly apply the existing calibration methods in supervised learning into crowdsourcing. On the other hand, the label distribution $\boldsymbol{P}_i$ is impacted by the noisy labels and the aggregated label $\hat{y}_i$ is determined by $\boldsymbol{P}_i$. These uncertainties and couplings increase the difficulty of confidence calibration in crowdsourcing. Additionally, there is another work worth comparing. Zong et al. (2024) directly apply calibration to learning from noisy labels, and their conclusion supports our claim that confidence calibration is essential for downstream tasks of label aggregation. However, the input provided for calibration in Zong et al. (2024) is $\hat{\mathcal{D}} = \{(\boldsymbol{x}_i, \hat{y}_i)\}_{i=1}^N$, which still fails to fully utilize the information in crowdsourcing.

# 4 THE PROPOSED METHOD

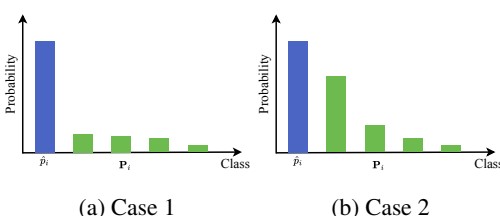

(a) Case 1      (b) Case 2

Figure 1: Underlying idea of LDLCC.

In this section, we provide a detailed description of our proposed LDLCC. Inspired by the label distribution learning, LDLCC tries to calibrate the confidence by learning the label distributions of all instances. Its underlying idea is illustrated in Figure 1. LDLCC divides instances into two cases based on their label distribution. When Case 1 shown in Figure 1(a) is satisfied, there is no other probability term in the label distribution $\boldsymbol{P}_i$ close to $\hat{p}_i$, indicating that the corresponding aggregated label has no confusing classes. At this point, the corresponding instance typically does not contain ambiguous attributes, and label aggregation should be more confident. Conversely, when Case 2 shown in Figure 1(b) is satisfied, there exist other probability terms in $\boldsymbol{P}_i$ close to $\hat{p}_i$, indicating that the corresponding aggregated label has confusing classes. At this point, the corresponding instance typically contains ambiguous attributes, and label aggregation should not be overly confident. LDLCC integrates the above analysis and calibrates confidence through two steps: label distribution refinement and label distribution learning.

## 4.1 LABEL DISTRIBUTION REFINEMENT

This step is primarily designed for high-confidence instances that satisfy Case 1. Considering that the aggregated label of high-confidence instances does not have confusing classes, the probability terms other than $\hat{p}_i$ in $\boldsymbol{P}_i$ should be 0. When non-zero probability terms appear, they are more likely to be caused by noisy labels. Therefore, LDLCC refines the label distributions of these high-confidence instances through sharpening to mitigate the impact of noisy labels. The key problem in this step is how to identify high-confidence instances that satisfy Case 1. Inspired by confident learning (Northcutt et al., 2021), LDLCC first calculates the average confidence $\mu_{c_q}$ for each class $c_q$ as follows:

$$\mu_{c_q} = \frac{\sum_{i=1}^N \mathbb{I}(\hat{y}_i = c_q)\hat{p}_i}{\sum_{i=1}^N \mathbb{I}(\hat{y}_i = c_q)}. \tag{4}$$

Then, LDLCC identifies high-confidence instances $\boldsymbol{X}_h$ that satisfy Case 1 as follows:

$$\boldsymbol{X}_h = \{\boldsymbol{x}_i \mid \hat{p}_i \geq \mu_{\hat{y}_i}, \text{ for } i = 1, 2, \ldots, N\}. \tag{5}$$

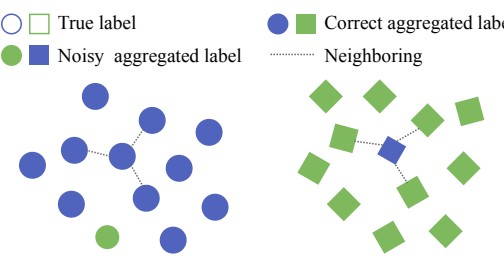

Figure 2: The illustration of eliminating falsely high-confidence instances using neighbors.

By this way, LDLCC filters out instances with confidence exceeding the average confidence in each class. However, revisiting the example we provided in the introduction using MV, apart from the aggregated labels not containing confusing classes, there is another scenario that could lead to high confidence. If an instance has received only a small number of crowd labels, the label probability $P_i$ may be unreliable, resulting in a high $\hat{p}_i$. Therefore, the current $X_h$ is not sufficiently convincing. To further eliminate those falsely high-confidence instances, LDLCC finds the neighbors for each instance in $X_h$ over the attribute space $\mathcal{X}$. The illustration of this process is shown in Figure 2. Falsely high-confidence instances, although achieving high confidence in the case of receiving only a small number of crowd labels, may have an aggregated label that differs from those of other neighbors in $X_h$. Therefore, LDLCC queries neighbors for each instance in $X_h$ and eliminates instance whose aggregated label differ from those of its neighbors.

According to the manifold hypothesis (Narayanan & Mitter, 2010), the local geometric structure of the data can be measured using Euclidean distance. Therefore, LDLCC does not make additional assumptions about $\mathcal{X}$ and directly calculates the Euclidean distance between each pair of instances $x_i$ and $x_j$ in $X_h$ as follows:

$$d_{ij} = \sqrt{\sum_{m=1}^{M} (x_{im} - x_{jm})^2}. \tag{6}$$

Equation (6) requires all attributes to be numerical, so we need to perform one-hot encoding on the nominal attributes before inputting $\hat{D}$ into LDLCC. Then, LDLCC sorts the distances and finds the $K$ nearest neighbors $\mathcal{N}_i$ for each instance $x_i$ in $X_h$. Subsequently, LDLCC compares the aggregated labels of $x_i$ and its neighbors $\mathcal{N}_i$ as follows:

$$s(x_i, \mathcal{N}_i) = \begin{cases} 1 & \text{if } \exists x_j \in \mathcal{N}_i \text{ such that } \hat{y}_i \neq \hat{y}_j \\ 0 & \text{otherwise} \end{cases}. \tag{7}$$

Here, $s(x_i, \mathcal{N}_i) = 1$ indicates that the aggregated label of $x_i$ differs from the aggregated labels of its neighbors in $X_h$. Therefore, LDLCC further updates $X_h$ as follows:

$$X_h = \{x_i \mid s(x_i, \mathcal{N}_i) = 0, \text{ for } i = 1, 2, \ldots, |X_h|\}. \tag{8}$$

Finally, LDLCC treats the instances in $X_h$ as high-confidence instances that satisfy Case 1. To mitigate the impact of noisy labels, LDLCC sharpens the label distribution $P_i = \{P_{iq}\}_{q=1}^{Q}$ of $x_i \in X_h$ as follows:

$$P_{iq} = \begin{cases} 1 & \text{if } \hat{y}_i = c_q \\ 0 & \text{otherwise} \end{cases}. \tag{9}$$

## 4.2 LABEL DISTRIBUTION LEARNING

This step primarily addresses the problem of how to calibrate the confidence of aggregated labels. Inspired by the label distribution learning (Xu & Geng, 2019; Lu et al., 2023), we argue that $P_i$ reflects the degree of membership of $x_i$ to each class. Based on this argument, even when $P_i$ satisfies Case 2, learning the mapping from the attribute space $\mathcal{X}$ to the confusing classes can effectively help $\hat{p}_i$ mitigate overconfidence. Therefore, LDLCC captures the mapping relationship from $\mathcal{X}$ to $\mathcal{Y}$ by label distribution learning.

According to Zong et al. (2024), one key reason for network overconfidence is the translation invariance of softmax. Therefore, as shown in Figure 3, LDLCC constructs a regression task instead of a classification task in label distribution learning to avoid using the softmax function. Specifically, LDLCC takes all instances in $\hat{D}$ as input and uses their label distributions as targets to train a regression network. If an instance is identified as high-confidence in the first step, LDLCC uses its

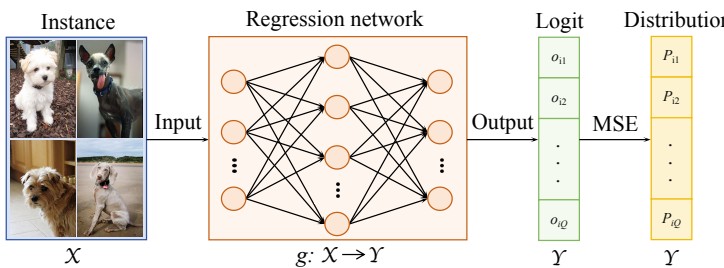

Figure 3: The illustration of label distribution learning.

refined label distribution; otherwise, it adopts the label distribution derived from label aggregation. The regression network $g : \mathcal{X} \to \mathcal{Y}$ is trained using the mean squared error (MSE) loss as follows:

$$\mathcal{L}_{\text{MSE}} = \frac{1}{N} \sum_{i=1}^{N} \| \boldsymbol{P}_i - \boldsymbol{o}_i \|_2^2, \tag{10}$$

where $\boldsymbol{o}_i$ is the logit of $\boldsymbol{x}_i$ output by $g$. Considering that $\boldsymbol{P}_i$ is impacted by noisy labels, $\boldsymbol{P}_i$ can be expressed as follows:

$$\boldsymbol{P}_i = \boldsymbol{P}_i^t + \boldsymbol{\epsilon}, \quad \boldsymbol{\epsilon} \sim \mathcal{N}(0, \sigma^2 \boldsymbol{I}), \tag{11}$$

where $\boldsymbol{P}_i^t$ is the true label distribution of $\boldsymbol{x}_i$ and $\boldsymbol{\epsilon}$ is the noise term. Then, the MSE loss can be derived as follows:

$$\begin{aligned}
\mathcal{L}_{\text{MSE}} &= \mathbb{E}\left[ \| \boldsymbol{P} - \boldsymbol{o} \|_2^2 \right] = \mathbb{E}\left[ \| \boldsymbol{P}^t + \boldsymbol{\epsilon} - \boldsymbol{o} \|_2^2 \right] \\
&= \mathbb{E}\left[ \| \boldsymbol{P}^t - \boldsymbol{o} \|_2^2 \right] + 2\mathbb{E}\left[ (\boldsymbol{P}^t - \boldsymbol{o})^T \boldsymbol{\epsilon} \right] + \mathbb{E}\left[ \| \boldsymbol{\epsilon} \|_2^2 \right]
\end{aligned} \tag{12}$$

Here, $\mathbb{E}\left[ \| \boldsymbol{\epsilon} \|_2^2 \right] = Q\sigma^2$. Because $\boldsymbol{\epsilon}$ is independent of $\boldsymbol{P}^t$ and $\boldsymbol{o}$ so that $\mathbb{E}\left[ (\boldsymbol{P}^t - \boldsymbol{o})^T \boldsymbol{\epsilon} \right] = 0$. Therefore, Equation (12) can be simplified as follows:

$$\mathcal{L}_{\text{MSE}} = \mathbb{E}\left[ \| \boldsymbol{P}^t - \boldsymbol{o} \|_2^2 \right] + Q\sigma^2. \tag{13}$$

From Equation (13), we can see that the effect of noise on the MSE loss is a fixed constant, which means that the MSE loss is relatively robust to noise. Therefore, the MSE loss can be used to learn the mapping relationship from $\mathcal{X}$ to $\mathcal{Y}$ as accurately as possible. Ultimately, LDLCC obtain the calibrated confidence of $\hat{y}_i$ as follows:

$$\tilde{p}_i = \max \frac{\boldsymbol{o}_i}{\| \boldsymbol{o}_i \|_1}. \tag{14}$$

In addition to the above details, due to the limited pages, the whole learning process of LDLCC and its time complexity analysis are provided in **Appendix A**.

## 5 EXPERIMENTS

In this paper, we define confidence calibration for crowdsourcing and propose the LDLCC method. Therefore, to validate the contributions of this paper, we need to answer the following questions:

- **Q1:** Do existing label aggregation methods suffer from miscalibration issues?
- **Q2:** Can LDLCC effectively calibrate the confidence for label aggregation methods?
- **Q3:** Is LDLCC better suited for crowdsourcing compared to existing calibration methods?
- **Q4:** Can LDLCC further improve the performance of downstream tasks?

This section presents our experimental setup, results, and analysis centered around these questions.

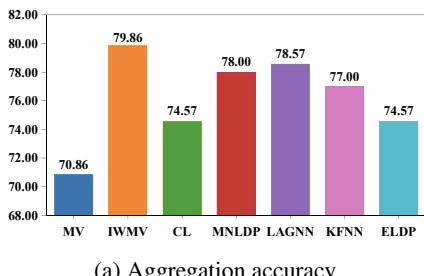
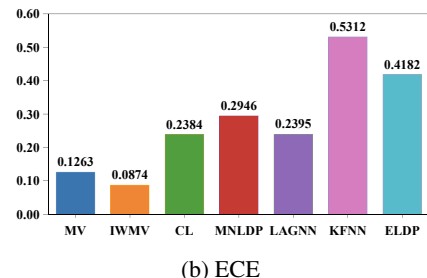

(a) Aggregation accuracy

(b) ECE

Figure 4: Aggregation accuracy (%) and ECE of MV, IWMV, CL, MNLDP, LAGNN, KFNN, and ELDP on *Music* dataset.

## 5.1 EXPERIMENTAL SETUP

**Datasets.** We conduct experiments on three widely-used real-world datasets: *Music*, *LabelMe*, and *Income*. All these datasets are collected through Amazon Mechanical Turk (AMT). Among them, *Music* contains 700 instances, 31 numeric attributes, and 10 classes. It is annotated by 44 workers, resulting in a total of 2946 crowd labels. *LabelMe* contains 1000 instances, 512 numeric attributes, and 8 classes. It is annotated by 59 workers, resulting in a total of 2547 crowd labels. *Income* contains 600 instances, 10 nominal attributes, and 2 classes. It is annotated by 67 workers, resulting in a total of 6000 crowd labels. Considering the requirements of Equation (6), before feeding the datasets into LDLCC, we apply numeric encoding to nominal attributes using scikit-learn's LabelEncoder, followed by standardizing all attributes with scikit-learn's StandardScaler.

**Baseline Methods.** The label aggregation methods used in our experiments include MV (Sheng et al., 2008), Iterative Weighted Majority Voting (IWMV) (Li & Yu, 2014), Crowd Layer (CL) (Rodrigues & Pereira, 2018), Multiple Noisy Label Distribution Propagation (MNLDP) (Jiang et al., 2022), Label Aggregation with Graph Neural Networks (LAGNN) (Ying et al., 2024), K-Free Nearest Neighbor (KFNN) (Zhang et al., 2024), and Enhanced Label Distribution Propagation (ELDP) (Zhang et al., 2025). For MV, we utilize the implementation provided by the Crowd Environment and its Knowledge Analysis (CEKA) platform (Zhang et al., 2015). The implementations of IWMV, MNLDP, LAGNN, KFNN, and ELDP are sourced from their respective authors. Both CL and our proposed LDLCC are implemented in Python. All parameter settings of baseline methods are consistent with those specified in their original papers. For our proposed LDLCC, we set the number of nearest neighbors $K = 3$. In addition, the regression network $g$ in LDLCC is implemented as a simple four-layer dense neural network. The first hidden layer consists of 64 units, while the second hidden layer has 128 units, both using the ReLU activation function. MSE is employed as the loss function, and the Adam optimizer with a learning rate of 0.001 is used for training. The network is trained for 1000 epochs. To ensure a fair comparison, the same architecture for $g$ are adopted as the backbone network for CL.

**Metrics.** To assess calibration performance, we use ECE as the evaluation metric in this paper, with the number of bins $T$ set to 10. To mitigate the effects of randomness in experiments, each method is executed 10 times on each dataset, and the average results are reported.

## 5.2 EXPERIMENTAL RESULTS.

**Experimental Results for Q1.** We compare the performance of each label aggregation method on each dataset in terms of aggregation accuracy and ECE. Here, aggregation accuracy is calculated as the ratio of the number of correctly aggregated labels to the total number of instances. ECE is calculated by Equation (3). Due to the limited pages, the results on the dataset *Music* are presented in Figure 4, while the results for the other two datasets are provided in **Appendix B**. As shown in Figure 4, compared to the simplest MV method, all more advanced methods achieve higher aggregation accuracy. However, except for IWMV, these advanced methods result in worse ECE. This observation suggests that while more advanced label aggregation methods enhance aggregation accuracy, they often degrade confidence calibration performance. These findings reveal that existing

Table 1: ECE comparisons of seven label aggregation methods before and after using our proposed LDLCC on three datasets.

| Dataset | MV | | IWMV | | CL | | MNLDP | | LAGNN | | KFNN | | ELDP | |
|---|---|---|---|---|---|---|---|---|---|---|---|---|---|---|
| | ORI | LDLCC | ORI | LDLCC | ORI | LDLCC | ORI | LDLCC | ORI | LDLCC | ORI | LDLCC | ORI | LDLCC |
| *Music* | 0.1263 | **0.1113** ● | 0.0874 | 0.0930 | 0.2384 | **0.0956** ● | 0.2946 | **0.2258** ● | 0.2395 | 0.2643 ○ | 0.5312 | **0.4192** ● | 0.4182 | **0.3146** ● |
| *LabelMe* | 0.1078 | **0.0756** ● | 0.0983 | **0.0678** ● | 0.2017 | **0.1173** ● | 0.1840 | **0.1193** ● | 0.2174 | **0.1851** ● | 0.3638 | **0.2113** ● | 0.2988 | **0.1388** ● |
| *Income* | 0.0402 | 0.0499 ○ | 0.0883 | **0.0776** ● | 0.2488 | **0.1099** ● | 0.0429 | **0.0402** | 0.1994 | **0.0961** ● | 0.0773 | **0.0755** | 0.0289 | 0.0374 ○ |
| Average | 0.0914 | **0.0790** | 0.0913 | **0.0795** | 0.2296 | **0.1076** | 0.1739 | **0.1284** | 0.2188 | **0.1818** | 0.3241 | **0.2353** | 0.2486 | **0.1636** |

label aggregation methods suffer from miscalibration issues, which highlights the importance and necessity of performing confidence calibration in our work.

**Experimental Results for Q2.** We compare the ECE performance of each label aggregation method before and after using LDLCC on all three datasets. The results are shown in Table 1. Here, the symbols ● and ○ in the table denote the ECE has a statistically significant improvement or degradation using LDLCC with a corrected paired two-tailed t-test with the significance level $\alpha$ = 0.05 (Nadeau & Bengio, 2003), respectively. From the results shown in Table 1, we can summarize the following highlights: i) On dataset *Music*, LDLCC reduces the ECE of all baseline methods except for IWMV and LAGNN. On dataset *LabelMe*, LDLCC reduces the ECE of all baseline methods. On dataset *Income*, LDLCC reduces the ECE of all baseline methods except for MV and ELDP. LDLCC significantly reduces the ECE in 15 cases, failing in 3 cases. ii) The average ECE of the baseline methods before using LDLCC is as follows: MV (0.0914), IWMV (0.0913), CL (0.2296), MNLDP (0.1739), LAGNN (0.2188), KFNN (0.3241), and ELDP (0.2486). After using LDLCC, the average ECE of the baseline methods decreases to MV (0.0790), IWMV (0.0795), CL (0.1076), MNLDP (0.1284), LAGNN (0.1818), KFNN (0.2353), and ELDP (0.1636). LDLCC effectively reduces the average ECE of all baseline methods. These experimental results validate the effectiveness of our LDLCC in calibrating the confidence for existing label aggregation methods.

**Experimental Results for Q3.** We compare the performance of our proposed LDLCC with existing calibration methods in supervised learning. As discussed above, existing calibration methods in supervised learning can be divided into three categories. We compare LDLCC with the most representative method from each category, and their specific details are as follows:

- For post-hoc calibration, we use Temperature Scaling (Guo et al., 2017) as the baseline and set the temperature parameter to 3.

- For regularization-based calibration, we use Label Smoothing (Müller et al., 2019) as the baseline and set the smoothing factor to 0.1.

- For loss-based calibration, we use Focal Loss (Mukhoti et al., 2020) as the baseline and set the focal factor to 3.

Here, for Temperature Scaling (TS), since there is no validation set, we empirically set the temperature parameter to 3 to avoid overconfidence in the results. For Label Smoothing (LS) and Focal Loss (FL), we use the suggested parameter settings in their original papers. For fairness, all these methods use the same backbone network $g$ from LDLCC. Except for FL, all methods adopt the cross-entropy loss as the training loss. As the true labels are unavailable, we use aggregated labels as the target labels. Based on these settings, we fix the label aggregation method as MV and the dataset as *Music* for the experiments. The results are shown in Figure 5. From it, we can observe that, apart from our proposed LDLCC, none of these calibration methods can further reduce the ECE of MV. These results indicate that our proposed LDLCC is more suitable for crowdsourcing compared to calibration methods in supervised learning.

**Experimental Results for Q4.** We implement the latest Confidence Learning-based Noise Correction (CLNC) (Su et al., 2026) method as the downstream task to verify the effectiveness of confidence calibration. CLNC is used to correct the aggregated labels of MV both before and after confidence calibration by LDLCC on dataset *Music*. The noise ratios of CLNC are shown in Figure 6. Here, the noise ratio is calculated as 1 minus the aggregation accuracy. From the results shown in Figure 6, it can be observed that, after confidence calibration by LDLCC, the noise ratio of MV

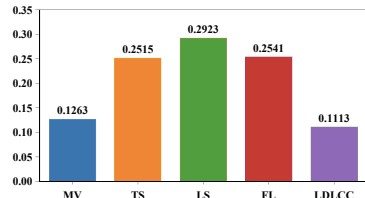 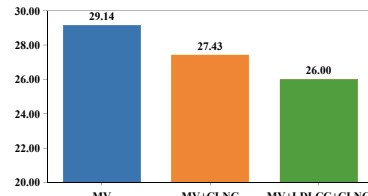

Figure 5: ECE comparisons of MV, TS, LS, FL, and LDLCC on *Music* dataset.

Figure 6: Noise ratio (%) comparisons before and after using LDLCC on *Music* dataset.

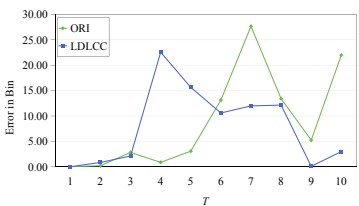 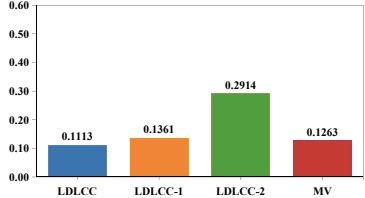

Figure 7: Total error comparisons in each bin of ECE on *Music* dataset.

Figure 8: ECE comparisons of LDLCC and its variants on *Music* dataset.

can be further reduced by CLNC. These results indicate that our LDLCC can effectively improve the performance of downstream tasks after label aggregation.

### 5.3 DISCUSSION AND ANALYSIS

From the answers to the four questions above, it is evident that the motivation for defining confidence calibration in crowdsourcing is justified, and the proposed LDLCC method is effective. Now, we provide further analysis to demonstrate other underlying characteristics of LDLCC.

**Calibration Analysis.** To observe the calibration behavior of LDLCC in a more fine-grained manner, we visualize the total error, defined as $|B_t|\|a_t - \bar{p}_t|$, in each bin $B_t$ of ECE. For the experiments, we still fix the label aggregation method as MV and the dataset as *Music*. The results are presented in Figure 7. It can be observed that LDLCC tends to prioritize calibrating bins with higher confidence. This indicates that, after calibration by LDLCC, higher calibrated confidence become more accurate. However, it is worth noting that Figure 7 also highlights a limitation of LDLCC: low confidence calibrated by LDLCC may be inaccurate.

**Ablation Study.** To investigate the effectiveness of the two steps in LDLCC, we conduct an ablation study based on MV and the *Music* dataset. Specifically, we implement two variants of LDLCC: LDLCC-1 removes the whole label distribution refinement step from LDLCC, and LDLCC-2 removes the whole label distribution learning step from LDLCC. The results are shown in Figure 8. It can be observed from Figure 8 that each step is crucial to the effectiveness of LDLCC. Removing either the label distribution refinement step or the label distribution learning step will degrade the calibration performance of LDLCC, making it perform worse than the original MV.

## 6 CONCLUSION

In this paper, we define the confidence calibration for crowdsourcing and propose a novel Label Distribution Learning-based Confidence Calibration (LDLCC) method. LDLCC identifies the high-confidence instances and refines their label distributions to mitigate the impact of noisy labels. Subsequently, LDLCC calibrates the confidence by label distribution learning. Experimental results demonstrate the effectiveness and other underlying characteristics of LDLCC.

However, as an initial method for confidence calibration in crowdsourcing, the current LDLCC still has some limitations. For example, it tends to prioritize calibrating bins with higher confidence. In the future, we will work toward further improving the performance of LDLCC in this direction.

## REPRODUCIBILITY STATEMENT

We submit the code and datasets as supplementary materials, and the details of dataset preprocessing and algorithm implementation are provided in the main text. Once our paper is accepted, we will make the code and datasets publicly available on GitHub.

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

# Appendix A    The whole learning process and time complexity analysis

---

**Algorithm 1** The learning process of LDLCC

---

**Input**: Aggregated dataset $\hat{\mathcal{D}} = \{\boldsymbol{x}_i, \boldsymbol{P}_i, \hat{y}_i, \hat{p}_i\}_{i=1}^N$
**Parameter**: The number of nearest neighbors $K$
**Output**: Calibrated dataset $\tilde{\mathcal{D}} = \{\boldsymbol{x}_i, \boldsymbol{P}_i, \hat{y}_i, \tilde{p}_i\}_{i=1}^N$

  1: **for** $q = 1$ to $Q$ **do**
  2:     Calculate $\mu_{c_q}$ for $c_q$ by Equation (4).
  3: **end for**
  4: Identify high-confidence instances $\boldsymbol{X}_h$ by Equation (5).
  5: **for** $i = 1$ to $|\boldsymbol{X}_h|$ **do**
  6:     **for** $j = 1$ to $|\boldsymbol{X}_h|$ **do**
  7:         Calculate $d_{ij}$ for $\boldsymbol{x}_i$ and $\boldsymbol{x}_j$ by Equation (6).
  8:     **end for**
  9:     Sort the distances and query $K$ neighbors $\mathcal{N}_i$ for $\boldsymbol{x}_i$.
 10:     Calculate $s(\boldsymbol{x}_i, \mathcal{N}_i)$ for $\boldsymbol{x}_i$ by Equation (7).
 11: **end for**
 12: **for** $i = 1$ to $|\boldsymbol{X}_h|$ **do**
 13:     **if** $s(\boldsymbol{x}_i, \mathcal{N}_i) = 0$ **then**
 14:         Sharpen $\boldsymbol{P}_i$ for $\boldsymbol{x}_i$ by Equation (9).
 15:     **else**
 16:         Remove $\boldsymbol{x}_i$ from $\boldsymbol{X}_h$.
 17:     **end if**
 18: **end for**
 19: Train the regression network $g$ by Equation (10).
 20: **for** $i = 1$ to $N$ **do**
 21:     Obtain $\tilde{p}_i$ for $\boldsymbol{x}_i$ by Equation (14).
 22: **end for**
 23: **return** $\tilde{\mathcal{D}} = \{\boldsymbol{x}_i, \boldsymbol{P}_i, \hat{y}_i, \tilde{p}_i\}_{i=1}^N$

---

In summary, the complete learning process of LDLCC is shown in Algorithm 1. In Algorithm 1, lines 1-3 calculate the average confidence $\mu_{c_q}$ for each class $c_q$ and their time complexity is $O(NQ)$. Line 4 identifies high-confidence instances $\boldsymbol{X}_h$ and its time complexity is $O(N)$. Lines 6-8 calculate the distances between $\boldsymbol{x}_i$ and each instance $\boldsymbol{x}_j$ in $\boldsymbol{X}_h$ and their time complexity is $O(NM)$. Line 9 sorts the distances and queries the neighbors $\mathcal{N}_i$ for $\boldsymbol{x}_i$ and its time complexity is $O(N \log N)$. Line 10 compares the aggregated labels of $\boldsymbol{x}_i$ and its neighbors $\mathcal{N}_i$ and its time complexity is $O(K)$. Due to $K \ll N$, the time complexity of lines 6-10 is $O(N(M + \log N))$. Therefore, the time complexity of lines 5-11 is $O(N^2(M + \log N))$. Lines 12-18 refine the label distribution for high-confidence instances and their time complexity is $O(NQ)$. Let $O(t_1)$ and $O(t_2)$ denote the training and test time complexity of $g$, respectively. Line 19 trains $g$ and its time complexity is $O(t_1)$. Line 20-22 obtain the calibrated confidence for each instance in $\hat{\mathcal{D}}$ and their time complexity is $O(N(t_2 + Q))$. Considering only the highest-order terms, the overall time complexity of LDLCC is $O(N^2(M + \log N) + t_1 + N(t_2 + Q))$.

## Appendix B    Experimental results on datasets *LabelMe* and *Income*

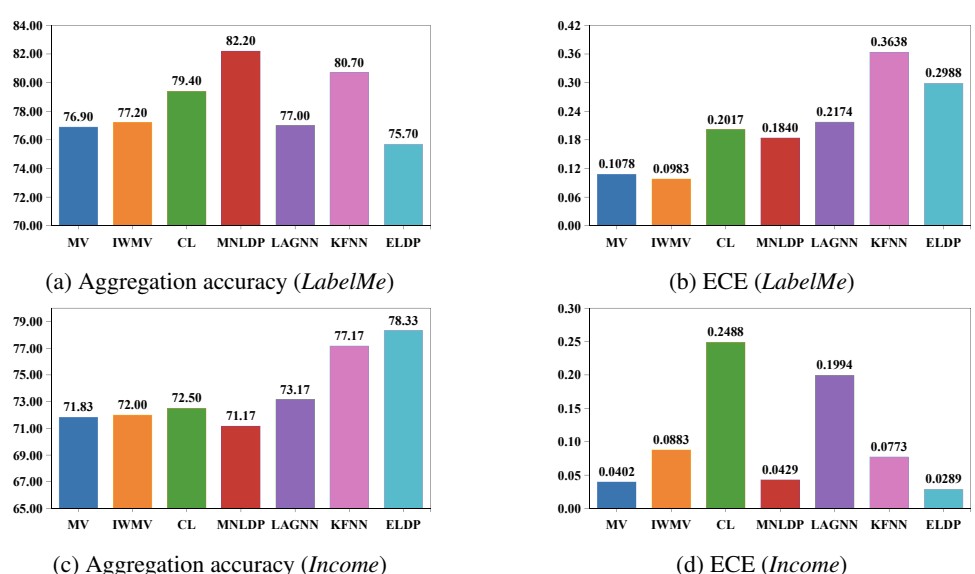

(a) Aggregation accuracy (*LabelMe*)

(b) ECE (*LabelMe*)

(c) Aggregation accuracy (*Income*)

(d) ECE (*Income*)

Figure 9: Aggregation accuracy (%) and ECE of MV, IWMV, CL, MNLDP, LAGNN, KFNN, and ELDP on datasets *LabelMe* and *Income*.

