# OpenReview forum: "LDLCC: Label Distribution Learning-based Confidence Calibration for Crowdsourcing"
_ICLR.cc/2026/Conference — ICLR 2026 Conference Withdrawn Submission_

### Official Review · Reviewer_KSTN · 2025-11-01

**Soundness:** 2
**Presentation:** 3
**Contribution:** 2
**Rating:** 2
**Confidence:** 4

**Summary:**

The paper tackles the issue of miscalibration in crowdsourced label aggregation. To address this, the authors formally define confidence calibration in the crowdsourcing setting and introduce a Label Distribution Learning-based Confidence Calibration (LDLCC) framework. Specifically,

- LDLCC first identifies high-confidence instances by combining confident learning with a $K$-nearest-neighbor consistency check, and refines their label distributions through a sharpening procedure.

- It then trains a regression network to learn calibrated label distributions by minimizing the mean squared error (MSE) between the predicted and refined distributions.

**Strengths:**

1. The paper studies the problem of miscalibration in the context of label aggregation for crowdsourcing.

2. The paper proposes LDLCC to address the problem.

3. The paper includes experimental results to validate the proposed method.

**Weaknesses:**

### Weaknesses, Detailed Comments, and Questions:

1. The mathematical formulation of calibration in Eqs. (1)–(2) is not rigorous. Specifically, the *probability* in Eq. (1) and the *expectation* in Eq. (2) are not defined with respect to any explicit probability measure or sample space. These expressions should be written explicitly to clarify the underlying stochastic model.

2. Weak connection to the crowdsourcing setup.

    - Although the paper claims to “formally define confidence calibration for crowdsourcing,” the definitions of *perfect calibration* (Eq. 1) and *Expected Calibration Error (ECE)* (Eq. 2) are mathematically identical to those used in standard supervised learning (Guo et al., 2017).

    - Furthermore, although Section 3 introduces the crowdsourced noisy labels $\mathbf{L}_i$, the subsequent sections, including the calibration formulation (Eqs. 1–2) and algorithm design (Sections 4.1–4.2, Algorithm 1), operate solely on the aggregated label distributions $\mathbf{P}_i$. The paper does not explain why the proposed framework is *specific* to crowdsourcing.

    - In fact, if the crowdsourcing context were removed, most of the analysis and algorithmic structure would remain unchanged. Thus, the method does not exploit key properties of crowdsourced data such as label sparsity, annotator heterogeneity, or instance-dependent noise.


3. In Section 4.2 (Eqs. 11–13), the authors assume additive Gaussian noise $\epsilon\sim\mathcal{N}(0,\sigma^2\mathbf{I})$ to argue that “the effect of noise on the MSE loss is a fixed constant, which means that the MSE loss is relatively robust to noise.” This assumption is unrealistic in the context of crowdsourced labels:

    - Real annotation noise is highly *instance- and annotator-dependent*, rather than i.i.d. Gaussian.

    - The variable $\mathbf{P}_i$ itself results from an aggregation process, not a direct observation, so the additive noise model is conceptually inconsistent.

    - Consequently, the claimed robustness of the MSE loss to noise lacks theoretical justification under the actual data-generating mechanism.

4. The three datasets (Music, LabelMe, Income) are small, and none represent modern large-scale crowdsourcing or complex perceptual labeling tasks (e.g., CIFAR-10H, AMT-based image classification).
    Moreover, several experiments, particularly those related to Q4 in Sec. 5.1 and Sec. 5.3, report results only on the Music dataset, without explaining why the other datasets were excluded.
    The experimental evidence therefore does not sufficiently support claims of robustness or generality.

5. While the paper targets an underexplored problem, the proposed LDLCC framework offers limited methodological novelty and does not contribute new theoretical insights or algorithmic principles beyond existing work.

**Questions:**

See above.

---

### Official Review · Reviewer_JwDK · 2025-11-01

**Soundness:** 1
**Presentation:** 1
**Contribution:** 2
**Rating:** 2
**Confidence:** 3

**Summary:**

This paper proposes LDLCC (Label Distribution Learning-based Confidence Calibration), a method designed to address the problem of confidence miscalibration in crowdsourced label aggregation. LDLCC introduces a two-stage framework—label refinement and label distribution learning—to improve calibration when ground-truth labels are unavailable. Experiments on multiple datasets and aggregation methods demonstrate improved calibration and downstream performance.

**Strengths:**

- The paper proposes a new and meaningful problem, extending calibration research to the crowdsourcing setting where true labels are not directly available.
- The authors design the LDLCC algorithm and verify its effectiveness across multiple benchmarks and label aggregation methods, including downstream validation, which enhances the practical relevance of the approach.

**Weaknesses:**

The algorithmic description is unclear and incomplete---the motivations for using specific techniques and key concepts are missing. For example, why is sample filtering and label refinement necessary? Why use the average confidence as a threshold? What is label distribution learning? Why does the translation invariance of softmax make the network overconfident? Besides, the validation for Q3 and Q4 is too limited, as it is only performed on the Music dataset with the MV method. More extensive experiments across datasets and aggregation methods would strengthen the reliability of the results.

**Questions:**

Please refer to the weaknesses.

---

### Official Review · Reviewer_4w4J · 2025-11-01

**Soundness:** 2
**Presentation:** 3
**Contribution:** 2
**Rating:** 2
**Confidence:** 3

**Summary:**

This paper focuses on the problem of confidence calibration in crowdsourcing. Traditional crowdsourcing label aggregation methods (such as MV, DS, LAGNN, etc.) primarily focus on label accuracy but ignore the inconsistency between the aggregation confidence and the label accuracy rate (i.e., miscalibration). The paper defines confidence calibration for crowdsourcing and proposes a Calibration method based on Label Distribution Learning (LDLCC, Label Distribution Learning-based Confidence Calibration).

**Strengths:**

1. Define the confidence calibration problem in the crowdsourcing scenario and point out its essential differences from supervised learning calibration
2. Introducing "Label Distribution Learning" into the calibration task is a novel idea
3. A two-stage strategy (sharpening + regression learning) is proposed, taking into account both noise robustness and distribution modeling capabilities
4. Not only evaluate ECE, but also verify the actual gains for downstream tasks; It includes complete ablation experiments to verify the necessity of each module

**Weaknesses:**

1.	Why not use temperature-scaled softmax or other calibration output layers? More thorough argumentation is needed.
2.	The actual crowdsourcing noise is non-Gaussian, non-independent, and category-dependent (such as some categories being easily confused). This assumption is too strong and weakens the theoretical support.
3.	Lack of comparison with the latest calibration methods: Although compared with TS/LS/FL, these are classic methods in supervised learning. In recent years, unsupervised/weakly supervised calibration methods (such as Dirichlet-based calibration, Zong et al., AAAI 2024 - this article has been cited but not used as a baseline) should be included in the comparison.
4.	The verification of downstream tasks is single: It was verified using only one noise correction method, CLNC, and was only demonstrated on the Music dataset. It should be extended to more downstream tasks (such as semi-supervised learning, robust training) and datasets.
5.	Computational overhead and scalability were not reported: The time complexity of LDLCC is O(N²(M + log N)) (Appendix A), which may not be feasible on large-scale data. However, the experiment did not discuss running time or memory usage.
6.	The literature review is somewhat insufficient: Although Zong et al. (2024) was mentioned, the essential differences between it and LDLCC were not discussed in depth (Zong uses the Dirichlet distribution to model and predict uncertainty, while LDLCC is based on regression learning label distribution). Recent works on crowdsourcing uncertainty modeling (such as Bayesian aggregation with uncertainty quantification) have not been cited.

**Questions:**

Same as weaknesses

---

### Note · Authors · 2025-11-22

I have read and agree with the venue's withdrawal policy on behalf of myself and my co-authors.